# The POU-Domain Transcription Factor Oct-6/POU3F1 as a Regulator of Cellular Response to Genotoxic Stress

**DOI:** 10.3390/cancers11060810

**Published:** 2019-06-11

**Authors:** Cinzia Fionda, Danilo Di Bona, Andrea Kosta, Helena Stabile, Angela Santoni, Marco Cippitelli

**Affiliations:** 1Department of Molecular Medicine, Sapienza University of Rome, 00161 Rome, Italy; andrea.kosta@uniroma1.it (A.K.); helena.stabile@uniroma1.it (H.S.); angela.santoni@uniroma1.it (A.S.); 2School and Chair of Allergology, Department of Emergencies and Organ Transplantation, University of Bari “Aldo Moro”, 70124 Bari, Italy; danilo.dibona@uniba.it; 3IRCCS, Neuromed, 86077 Pozzilli, Italy; 4Istituto Pasteur-Fondazione Cenci Bolognetti, Sapienza University of Rome, 00161 Rome, Italy

**Keywords:** DNA damage, ROS, Oct-6, Oct-1, chemotherapy

## Abstract

DNA damage and the generation of reactive oxygen species (ROS) are key mechanisms of apoptotic cell death by commonly used genotoxic drugs. However, the complex cellular response to these pharmacologic agents remains yet to be fully characterized. Several studies have described the role of transcription factor octamer-1 (Oct-1)/Pit-1, Oct-1/2, and Unc-86 shared domain class 2 homeobox 1 (POU2F1) in the regulation of the genes important for cellular response to genotoxic stress. Evaluating the possible involvement of other POU family transcription factors in these pathways, we revealed the inducible expression of Oct-6/POU3F1, a regulator of neural morphogenesis and epidermal differentiation, in cancer cells by genotoxic drugs. The induction of Oct-6 occurs at the transcriptional level via reactive oxygen species (ROS) and ataxia telangiectasia mutated- and Rad3-related (ATR)-dependent mechanisms, but in a p53 independent manner. Moreover, we provide evidence that Oct-6 may play a role in the regulation of cellular response to DNA damaging agents. Indeed, by using the shRNA approach, we demonstrate that in doxorubicin-treated H460 non-small-cell lung carcinoma (NSCLC) cells, Oct-6 depletion leads to a reduced G2-cell cycle arrest and senescence, but also to increased levels of intracellular ROS and DNA damage. In addition, we could identify p21 and catalase as Oct-6 target genes possibly mediating these effects. These results demonstrate that Oct-6 is expressed in cancer cells after genotoxic stress, and suggests its possible role in the control of ROS, DNA damage response (DDR), and senescence.

## 1. Introduction

DNA damaging agents in the form of γ-radiation or chemotherapeutic drugs are commonly used to treat cancer. For this reason, many efforts have been devoted to understanding the intricate mechanisms driving cellular survival and death upon DNA damage. In response to genotoxic stress, cells activate a complex signaling program, termed DNA damage response (DDR), leading to cell-cycle arrest, cellular senescence, and/or cell death to allow for DNA repair and to prevent the proliferation of damaged cells [1,2,3]. A central role in coordinating the DDR is played by the protein kinases ataxia telangiectasia mutated (ATM) and ATM- and Rad3-related (ATR) [4,5,6]. Following the recognition of DNA lesions by sensor proteins, ATM/ATR activate target checkpoint kinases (Chk1/Chk2), which are able to phosphorylate and modulate the activity of many downstream effectors, such as the protein p53, a key regulator of cellular stress response. Prominent p53 targets include important cell cycle modulating genes, such as p21 [7,8]. This protein blocks cell cycle progression, resulting in a temporary or permanent cell cycle arrest (senescence), and inhibits apoptosis [9,10].

A number of studies have demonstrated that, in response to DNA damage, the p21 gene expression can also be driven by other transcription factors, such as the Pit-1, Oct-1/2, and Unc-86 shared domain (POU) transcription factor family member octamer-1 (Oct-1) [11,12]. In particular, this transcription factor emerged as an important contributor to cell survival in response to various types of DNA damage; indeed, Oct-1-deficient cells have been described as hypersensitive to γ-radiation, doxorubicin (Dox), or hydrogen peroxide (H_2_O_2_), with increased levels of intracellular reactive oxygen species (ROS) [12,13]. In this regard, an additional Oct-1 target gene is catalase [14], an antioxidant enzyme that catalyzes the degradation of H_2_O_2_ and plays a crucial role in protecting cells against reactive oxygen species (ROS). These mechanisms of detoxification are relevant in the cells exposed to antitumor chemotherapeutics; indeed, these drugs are able to generate ROS production by causing DNA damage and apoptotic cell death.

In the present study, we described, for the first time, the increased expression of the transcription factor Oct-6 (POU3F1—POU class 3 homeobox 1 (an Octamer binding- POU-homeodomain family member)), a pivotal regulator of neural morphogenesis and epidermal differentiation [15,16,17,18,19], in various cancer cell lines following treatment with the chemotherapeutic drug Dox. Moreover, we characterized the molecular mechanisms required for Oct-6 induction, and defined a role for this transcription factor in the control of DNA damage-mediated stress response. Indeed, we found that in H460 NSCLC cells, the Dox-induced Oct-6 DNA binding activity is significantly reduced by ROS scavengers or by ATR inhibition via caffeine or shRNA silencing, but it is not affected by pifithrin-α, a pharmacological inhibitor of p53 protein. Moreover, we demonstrated that in Dox-treated cells, Oct-6 knockdown significantly reduces the proportion of G2/M phase and senescent cells, and increases the number of sub-G1 cells; concomitantly, it leads to higher intracellular ROS production and DNA damage. Interestingly, we also observed that Oct-6 can share with the POU domain protein Oct-1 the capability to regulate the common target genes required for cellular response to genotoxic stress, and possibly mediating these effects, such as p21 and catalase.

Overall, our data demonstrate that the transcription factor Oct-6 is induced by genotoxic stress in cancer cells, and may be an important player in the control of cellular events triggered by DNA damage.

## 2. Results

### 2.1. The Genotoxic Drug Doxorubicin Induces the Expression of Oct-6 in Human Cancer Cell Lines

To investigate regulatory pathways involved in the expression and activity of the transcription factors relevant in stress conditions, we analyzed the ability of the genotoxic drug Dox to modulate the expression and DNA binding activity of octamer-binding transcription factors in human cancer cell lines, given the role of Oct-1 as a stress sensor and regulator of cell survival in response to various types of stresses, including DNA damage [12,13]. Nuclear extracts were prepared from H460 NSCLC cells treated with 0.5 μM Dox for different lengths of time, and the DNA binding activity for octamer-binding factors was evaluated by electrophoretic mobility shift assay (EMSA), using a canonical probe.

As shown in Figure 1A, inducible octamer binding complexes were detected after 6 h, with a faster migration compared to constitutive binding complexes, and reached a maximum level after 9–12 h following Dox treatment. A supershift analysis confirmed the DNA binding of Oct-1 (Figure 1B), and, surprisingly, revealed a strong induction of Oct-6 containing complexes (Figure 1C), an Octamer binding the POU-homeodomain family member involved in the regulation of neural morphogenesis and epidermal differentiation [15,16,17,18]. Interestingly, as shown in Appendix A, a significant induction of octamer binding complexes with electrophoretic mobility similar to Oct-6 (as observed in the H460 cells used in these EMSAs as a control) was also detected in other human cancer cell lines of different tissue origin, suggesting that the Dox-mediated induction of Oct-6 was not restricted to the H460 NSCLC cell line. Importantly, these data were confirmed using the interactive NCI Transcriptional Pharmacodynamics Workbench (NCI TPW), which allows for the exploration of gene expression and transcriptional response to different anticancer agents across a well characterized ensemble of cancer cell lines (https://tpwb.nci.nih.gov/GeneExpressionNCI60/index.html) [20]; indeed, as shown in Appendix A, a significant induction over the basal expression of Oct-6 was detected in different cancer cell lineages treated with Dox (e.g., lung, melanoma, renal, and glioblastoma). The effect was particularly evident with Dox (and to a lesser extent with Topotecan, another topoisomerase inhibitor) in comparison with other chemotherapics (e.g., Sorafenib, Paclitaxel, Vorinostat, and Gemcitabine) in selected cell lines (Appendix A), and was confirmed in the GEO2R analysis of the microarray public dataset GSE116441 (available at http://www.ncbi.nlm.nih.gov/geo/; Appendix A) [20]. We further characterized the Oct-6 containing complexes using EMSA, and found that a similar binding pattern was also obtained in the H460 cells infected with a retrovirus vector encoding the full-length Oct-6 in the absence of Dox, suggesting a possible action of constitutive intracellular proteases able to generate shorter Oct-6 DNA binding fragments (Appendix A). Accordingly, the formation of faster migrating complexes was completely abolished by the calpain/cathepsin inhibitors E64 or ALLM (Calpain Inhibitor II; Appendix A), suggesting a constitutive post-transcriptional action of these proteases on Oct-6. Moreover, the pre-treatment of cells with the pan-caspase inhibitor Z-VAD (pan-caspase inhibitor) before exposure to Dox did not alter Oct-6 binding, excluding the fact that the faster migrating complexes are caused by toxic/apoptotic effects of the drug (Appendix A). 

### 2.2. Expression of Oct-6 Is Induced by Genotoxic Stress: Role of DDR, ATR, and ROS Production

We investigated the molecular mechanism(s) involved in the induction of Oct-6 by Dox. First, we observed that drug-induced Oct-6 DNA binding was blocked by the RNA and protein synthesis inhibitors actinomycin D and cycloheximide (Figure 2A). Then, to confirm that the up-regulation of Oct-6 could be associated with an increased mRNA expression, the total RNA was isolated from the H460 cells exposed to Dox for 24 h, and analyzed by real-time qRT-PCR. As shown in Figure 2B, we found a strong induction of Oct-6 mRNA expression in the Dox-treated cells. We also extended our analysis to other genotoxic drugs endowed with different mechanisms of action, and observed that mitomycin C (MMC), cisplatin (CPT), and etoposide (Eto) were all able to mediate the same effect as those observed with Dox (Appendix A), although to a different extent. These results confirm that genotoxic stress is able to induce Oct-6 mRNA expression in H460 NSCLC cells.

Genotoxic drugs, including Dox, have the capability of inducing ROS production, as well as ATM/ATR and p53 activation [21]. We thus evaluated the possible role of ROS generation by Dox in the induction of Oct-6 expression/DNA binding. H460 cells were pretreated with different concentrations of the antioxidants N-acetylcysteine (NAC; 10 to 60 mM) or pyrrolidine dithiocarbamate (PDTC; 100 to 500 µM), and then incubated with Dox for 24 h. As shown in Figure 2C, we found that the Oct-6 expression was significantly inhibited by NAC and PDTC.

Then, we tested whether caffeine, a widely used inhibitor capable of blocking both ATM and ATR catalytic activity [22], or pifithrin-α, an inhibitor of p53 activity [23], could interfere with the induction of the Oct-6 DNA binding activity in Dox-treated H460 cells.

To this purpose, the cells were pretreated with caffeine (from 1 to 2.5 mM) or with pifithrin-α (30 μM), and then incubated with Dox for 24 h. As shown in Figure 2D, we found that the Oct-6 expression was inhibited by caffeine, but not by pifithrin-α, suggesting that the activation of ATM/ATR is required for Oct-6 expression, and that p53 is not involved in this regulation. As a control, treatment with pifithrin-α could significantly revert the activity of Dox on cell-cycle arrest in G2 (Appendix A).

Based on these observations, the role of ATM/ATR kinases in the induction of Oct-6 expression was further investigated using shRNA approaches. As shown in Figure 2E,F, we observed that ATR, but not ATM silencing, can significantly reduce Oct-6 DNA binding activity by Dox.

Taken together, these results indicate that genotoxic stress induces the Oct-6 expression and DNA binding activity in cancer cells via the generation of ROS and DDR/ATR activation-dependent mechanisms.

### 2.3. Oct-6 as a Regulator of Drug-Induced Cell Stress Response

After Dox treatment, the cells enter a sustained arrest in the G2 phase of the cell cycle, and acquire a senescent phenotype.

To evaluate the possible role of Oct-6 in the regulation of cellular response to genotoxic stress, we assessed the impact of Oct-6 depletion on Dox-induced cell cycle checkpoint. We observed that, compared with non-targeting shRNA-infected cells, Dox-treated Oct-6/shRNA-transduced H460 cells (Figure 3A,B) display a significant lower proportion of G2/M phase cells, but also a higher number of apoptotic sub-G1 cells (Figure 3C,D), suggesting an effect of this induced transcription factor on the cell cycle and apoptosis in response to genotoxic stress.

We therefore extended our analysis to senescence, a cellular response associated to permanent G2 cell phase arrest in Dox-exposed cells, and to intracellular ROS production and oxidative damage in the cells treated with Dox.

Consistent with the observation of a reduced number of G2 phase-arrested cells, we found lower levels of cellular senescence, as evaluated by fluorescence-activated cell sorting (FACS) analysis of C_12_FDG fluorescence induced by intracellular senescence-associated *β*-galactosidase (SA-*β*Gal) activity, following treatment with Dox (Figure 4A). Moreover, we observed that the loss of Oct-6 expression led to a higher ROS production by Dox, as detected by the flow cytometry in the presence of the redox-sensitive dye dichloro-dihydro-fluorescein diacetate (DCFH-DA) (Figure 4B). On the contrary, the induction of ROS by Dox was significantly reduced in the Oct-6 overexpressing cells, suggesting a possible role for this transcription factor in controlling drug-induced oxidative stress (Figure 4C). Based on these observations, we hypothesized that the elevated ROS levels harbored by Oct-6 knockdown could lead to an increase in DNA damage. Consistently, after exposure to Dox, a higher percentage of cells expressing phosphorylated H2A histone family member X (γ-H2AX) was found in the Oct-6 silenced H460 cells as compared to the control cells (Figure 4D). 

Overall, these results indicate that Oct-6 may play an important role in the regulation of the critical cellular processes associated to drug-induced genotoxic stress.

### 2.4. Identification of Stress-Related Oct-6 Target Genes in Dox-Treated Cells

To identify the Oct-6 target gene(s) mediating the above described effects on Dox-induced DDR, we focused our attention on two proteins regulated by the homologous transcription factor Oct-1 in stressed cells, namely: p21 (CDKN1A), a critical regulator of cell cycle and senescence [11], and catalase, an important enzyme of ROS detoxification [14]. To this aim, we evaluated the impact of Oct-6 silencing on drug-induced p21 and catalase gene expression. We obtained the total RNA from non-targeting shRNA or Oct-6 shRNA infected H460 cells exposed to Dox for 24 h, and performed real-time qRT-PCR assays. Interestingly, we found that the Dox-induced p21 and catalase mRNA expression compared to the basal levels was significantly reduced in the H460 cells depleted of Oct-6 (Figure 5A); conversely, the expression levels of these mRNAs were all increased by ectopic Oct-6 over-expression (Figure 5B), suggesting a more general regulation of these genes by the Oct transcription factors in response to genotoxic stress. In this context, to evaluate the capability of Oct-6 to regulate the promoter activity of these genes, we performed transient co-transfection assays. As shown in Figure 5C, we observed a significant increase of p21 and catalase basal promoter activity in the untreated cells by Oct-6 overexpression. Accordingly, we also detected Oct-6 binding to known octamer consensus sequences in the context of the p21 promoter (Appendix A).

These results indicate that Oct-6 shares with the homologous member of POU-domain transcription factor Oct-1 the ability to target p21 and catalase genes in genotoxic stressed cells.

## 3. Discussion

Oct-6 is a class III POU domain protein that has been characterized primarily with respect to its expression in oligodendrocyte precursors, and in developing Schwann cells, where it is required for correctly timed transition to the myelinating state [15,24]. Moreover, it is also expressed in testes, keratinocytes, gastrointestinal epithelium, pancreatic β-cells, and macrophages, this suggesting a more general transcriptional activity for this protein [16,18,25,26,27,28,29,30]. However, the possible role of Oct-6 in cancer has never been investigated.

In this study, we describe the novel finding that the expression of Oct-6 is significantly induced after the treatment of different cancer cell lines with genotoxic drugs, suggesting that this protein may have a role in the cellular response to genotoxic stress. We found that the Oct-6 DNA binding activity is considerably inhibited by ROS scavengers, ATM/ATR inhibitor caffeine, and ATR shRNA interference, indicating that the ATR kinase activity together ROS generation are two absolute requirements for this induction.

Little information is available about the molecular mechanisms that regulate *Oct-6* gene expression. There is evidence that the expression of the Oct-6 protein is strongly enhanced by elevation of intracellular cAMP in Schwann cells [31]. Moreover, the *Oct-6* promoter is regulated by the Protein Kinase C (PKC) activator 12-O-Tetradecanoylphorbol-13-acetate (TPA) in cooperation with an estrogen-dependent enhancer, which might account for the expression difference of this gene between males and females [32]. A signal transducer and activator of transcription (STAT)1-binding region was also identified in the Oct-6 promoter, and the Oct-6 expression was observed in fibroblasts and macrophages in response to type I and type II interferon during viral infections [30]. A recent study has demonstrated a post-translational modification of the Oct-6 protein, which can be ubiquitinated and degraded upon SIRT1-dependent acetylation in stressed embryonic stem cells [33]. In the context of genotoxic stress, our observations indicate that p53 activation is not involved in the upregulation of Oct-6, at least in our experimental setting, and further experiments will be necessary to clarify the regulatory mechanisms underlying the transcriptional upregulation of this gene induced by genotoxic drugs and ROS. 

In the last years, much effort has been focused on understanding how cells respond to DNA damage and preserve genomic and chromatin integrity. A number of key cellular processes (e.g., growth arrest, DNA repair, or apoptosis) are implicated when cells are exposed to DNA damaging agents, and the specific inactivation or altered function of these pathways may frequently result in severe genomic instability (reviewed in the literature [34] and the references cited therein). 

The regulation of transcription factor genes and their activity after DNA damage is a crucial initial step, as their action can direct the transcription of the specific downstream effectors involved in the DNA damage response. In this regard, different observations have described the altered expression and/or activity of the Oct-1 protein in response to cellular stresses, such as DNA damage or serum withdrawal [12,35,36,37]. Indeed, the transcription factor Oct-1 has also been proposed as an important sensor and regulator of intracellular ROS production to maintain cellular integrity [12,13]. 

Our results indicate that Oct-6 may represent an additional POU domain transcription factor involved in the cellular response to stress. 

Intriguingly, following exposure to Dox, the H460 cells lacking Oct-6 expression show a reduced number of G2-cell cycle arrested and senescent cells, and a higher proportion of sub-G1 cells, suggesting the possible regulation by this transcription factor of the proteins required for G2/M block and the protection of cells from apoptosis. Consistently, upon exposure to Dox, Oct-6 silenced cells show higher levels of phosphorylated γH2AX, indicating increased levels of damaged DNA, possibly as a consequence of reduced cell cycle arrest and DNA repair; furthermore, a higher ROS production was also detected in the cells lacking Oct-6 in response to Dox that may enhance DNA damage and apoptosis (Figure 4). 

The ability of DNA damaging agents to trigger the G2/M cell cycle checkpoint to allow DNA repair and to inhibit apoptosis is caused, at least in part, by p53-dependent or independent p21 activation [38,39,40,41,42]; moreover, p21 is an essential mediator of cellular senescence [10].

Interestingly, our observations indicate that Oct-6 functions as a positive regulator of the p21 gene expression (Figure 5). Consistent with the data about the regulation of this gene by Oct-1 after ionizing radiation (IR) [11], our results indicate that Oct-6 binds to octamer consensus sequences of p21 promoter (Appendix A), and stimulates its activity (Figure 5C), indicating a more general regulation of this gene by octamer transcription factors. Accordingly, a significant reduction of p21 mRNA expression levels was revealed in drug-treated Oct-6 silenced H460 cells. 

Our results indicate that Oct-6 may function as a stress response effector in selected cancer cell lineages, and that the modification of cell cycle and senescence observed in cells lacking this transcription factor may be due to a decreased expression of p21.

As previously described for the transcription factor Oct-4 [13], Oct-6 may share with Oct-1 the capability to regulate the expression of common genes, specifically under stressful conditions. The convergence of homologous factors in the regulation of the same cellular targets may be a way to produce an appropriate response and to assure cell survival. Although Oct-1 is a constitutive factor, the kinetic of expression of the Oct-6 protein in drug-treated cells (appearing from 6 h after Dox treatment) lets us suppose a different timing of gene regulation by these transcription factors with a later role of Oct-6. These mechanisms may be required in the case of a compromised Oct-1 expression/function. Indeed, a prolonged exposure to ROS was shown to decrease Oct-1 expression, because of the methylation of the Oct-1 promoter [14].

Studies in a variety of cell types have suggested that ROS formation may be directly related to the ability of DNA damaging agents to induce apoptosis. In this regard, an additional functional consequence of the altered Oct-6 expression is the modification of genotoxic stress-induced ROS production. Indeed, the H460 cells lacking Oct-6 expression show a more elevated oxidative level (Figure 4B), whereas Oct-6 overexpression results in a decrease in the ROS levels in response to Dox (Figure 4C). These findings indicate not only that the Oct-6 induction requires ROS generation, but also that it is involved in oxidative cellular response, and may function as a rheostatic factor. In this regard, catalase, an antioxidant molecule protecting cells against an excess production of ROS, emerged in this study as a novel Oct-6 target gene (Figure 5). We detected a strong upregulation of catalase gene expression in H460 cells upon drug exposure. Such increased catalase expression levels may detoxify from ROS and inhibit a disproportional increase in oxidative levels, causing cell death [43,44]. Accordingly, targeting the redox status of cancer cells by modulating the catalase expression is emerging as a novel approach to potentiate chemotherapy [45]. Notably, as a positive regulator of the catalase gene expression, Oct-6 may contribute to this protective mechanism; in this context, further studies are needed to better define the role of this transcription factor in the modulation of ROS levels in stressed cells, possibly using CRISPR/Cas9 edited cell lines with the Oct-6 gene deleted, and mice models to test the efficacy of genotoxic drugs.

In conclusion, these data might help to gain a better understanding of the complex range of actions mediated by DNA damage response, and reveal a possible role of Oct-6 in the protective response of cells to genotoxic oxidative stress (Figure 6).

## 4. Materials and Methods 

### 4.1. Cell Lines and Reagents

The human tumor cell lines H460 (NSCLC), ZR-75 (breast), LnCaP (prostate), LoVo (colon), C33a (cervical), OVCA-433 (ovarian), EJ (bladder), RT112 (bladder), RT4 (bladder), Mel-120 (melanoma), SK-N-AS (neuroblastoma), and SHEP (neuroblastoma) were maintained at 37 °C and 5% CO_2_ in RPMI 1640 supplemented with 10% Foetal Calf Serum (FCS).

Doxorubicin, mitomycin C, cisplatin, etoposide, caffeine, N-acetylcysteine (NAC), pyrrolidine dithiocarbamate (PDTC), puromycin, G418, actinomycin D, cycloheximide, ALLM (Calpain Inhibitor II), E64, Pifithrin-α, and Z-VAD (pan-caspase inhibitor) were purchased from Sigma-Aldrich Chemical Co. (St. Louis, MO, USA). 

### 4.2. Flow Cytometric Analysis of Intracellular ROS

The intracellular ROS levels were measured by flow cytometry in the H460 cells loaded with the redox-sensitive dye DCFH-DA (Molecular Probes, Invitrogen, San Diego, CA, USA), after 48 h of treatment with 0.5 µM of Doxorubicin. The non-fluorescent DCFH-DA readily diffuses into the cells, where it is hydrolyzed to the polar derivative DCFH, which is oxidized in the presence of H_2_O_2_ to highly fluorescent DCF. The cells were incubated in the dark for 30 min at 37 °C with 10 µM DCFH-DA in phosphate-buffered saline (PBS), were harvested, and resuspended in PBS without DCFH-DA. The cells were analyzed using a FACSCanto (BD Bioscience, San Jose, CA, USA) and FlowJo (Version 7.2.5) Flow Cytometric Data Analysis Software (Tree Star, Inc., Ashland, OR, USA).

### 4.3. Plasmids

The expression vector for the murine Oct-6 gene (homology of 98.8% at the amino acid level with the human Oct-6) pCMV5-Oct-6 was kindly provided by Dr. M. Wegner (Universität Erlangen-Nürnberg, Erlangen, Germany). To generate the retroviral vector for Oct-6 overexpression, the cDNA encoding Oct-6 gene was cloned in pMSCV-Neo retroviral vector (Clontech, Mountain View, CA). For knocking down the human Oct-6 gene expression, we used a pLKO-1-sh-Oct-6 (SHCLND-NM_002699-TRCN0000020879) lentivirus vector with puromycin resistance, and the control vector pLKO non-targeting shRNA (MISSION™ Sigma-Aldrich, St. Louis MO, USA). For knocking down the human ATR gene and ATM expression, we used retroviral vectors pMSCV-sh-ATR or ATM, kindly provided by Dr. Ferbeyre G. (Université de Montréal, Canada) [46]. The pGL3-2320/+41 p21-promoter and pGL3-1518/+16-human catalase-promoter vector were kindly provided by Dr. Nenoi Mitsuru [21,22] (National Institute of Quantum and Radiological Sciences and Technology, Japan).

### 4.4. Transient Transfections

Log phase H460 cells were co-transfected with 5 µg of indicated luciferase reporter construct (pGL3-p21, or -catalase promoter or pGL3-basic) plus 2.5 µg of empty CMV-5 expression vector or encoding Oct-6. An RSV-gal or a TK-Renilla expression vector were co-transfected to normalize DNA uptake. The cell suspension mixed with DNA was kept on ice for 5 min, and electroporated in 0.45-cm electroporation cuvettes at 280 V, 960 μF, with a Gene Pulser apparatus (Bio-Rad, Richmond, CA, USA). Electroporated cells were diluted in a complete medium and plated on tissue-culture dishes. After 48 h, the cells were harvested, and the protein extracts were prepared for the luciferase and beta-galactosidase or renilla assays, as already described in the literature [47]. The protein concentration was quantified by the bicinchoninic acid assay (BCA) method (Pierce, Rockford, IL, USA). Luciferase activity was read using the luciferase reporter assay and the Glomax Multi Detection System (Promega Corp., Madison, WI), following the manufacturer’s instructions.

### 4.5. Virus Production and In Vitro Transduction

For the retrovirus production, phoenix cells were transfected with 5 μg of viral DNA using Lipofectamine Plus (Life Technologies, San Diego, CA, USA). The lentiviral vectors were co-transfected together with the packaging vectors pVSVG and psPAX2 into 293T cells using Lipofectamine Plus. Then, 48 h later, virus-containing supernatants were harvested, filtered, and used immediately for infections. The infections were performed on H460 cells in a complete medium with polybrene (8 μg/mL; hexadimethrine bromide; Sigma, St. Louis, MO, USA). After infection, the cells were allowed to expand for 24 h and were then selected for G418 (0.75 mg/mL) or puromycin (1 μg/mL) resistance. 

### 4.6. Electrophoretic Mobility-Shift Assay (EMSA)

The nuclear proteins were prepared as described in the literature [48]. The protein concentration of the extracts was determined by the BCA method (Pierce, Rockford, IL, USA). The nuclear proteins (10 μg) were incubated with radiolabeled DNA probes in a 20 µL reaction mixture containing 20 mM Tris (pH 7.5), 60 mM KCl, 2 mM EDTA, 0.5 mM dithiothreitol (DTT), 1–2 μg of poly(dI-dC), and 4% ficoll. Where indicated, a molar excess of double-strand oligomer was added as a cold competitor, and the mixture was incubated at room temperature for 10 min prior to adding the DNA probe. The nucleoprotein complexes were resolved as described in the literature [49]. Oligonucleotides were purchased by Eurofins Genomics (Ebersberg, Germany). Complementary strands were annealed and end-labeled as described in the literature [50]. Approximately 3 × 10^4^ cpm of labeled DNA was used in a standard electrophoretic mobility-shift assay reaction. In the supershift analysis, the specific antibody was added to the binding reaction, and the mixture was incubated for 30 min at room temperature prior to adding the labeled DNA probe. The antibody against Oct-1 was purchased from Santa Cruz Biotechnology (Santa Cruz, CA, USA). The antibody against Oct-6 was kindly provided by Dr. M. Wegner (Universität Erlangen-Nürnberg, Erlangen, Germany). The following double-strand oligomers were used as specific labeled probes or cold competitors (sense strand): hH2B Oct (Octamer-human-histone H2b), 5’-agctcttcaccttatttgcataagcgat-3’.

### 4.7. Western-Blot Analysis

For the Western-Blot analysis, nuclear proteins were prepared as described above [51]. The protein concentration of the nuclear and whole cell extracts was determined by the BCA method (Pierce, Rockford, IL, USA). From this, 30 to 50 μg of nuclear extract or whole cell extract were run on 10% denaturing Sodium Dodecyl Sulphate (SDS)-polyacrylamide gels. The proteins were then electroblotted onto nitrocellulose membranes (Schleicher and Schuell, Keene, NJ, USA) and blocked in 3% milk in Tris Buffered Saline Buffer with Tween (TBST). The immunoreactive bands were visualized on the nitrocellulose membranes, using horseradish-peroxidase-coupled goat anti-mouse immunoglobulins and the ECL detection system (GE Healthcare, Buckinghamshire, UK), following the manufacturer’s instructions. Antibodies against Hsp70, ATM, and ATR were purchased from Santa Cruz Biotechnology (Santa Cruz, CA, USA). Antibody against β–actin was purchased from Merk Millipore (Darmstadt, Germany) and Sigma-Aldrich (St. Louis, MO, USA), respectively.

### 4.8. RNA Isolation, RT-PCR, and Real-Time PCR

The total RNA was extracted using TRIZOL^TM^ (Thermo Fisher Scientific, Waltham, MA USA), according to manufacturer’s instructions. The concentration and quality of the extracted total RNA was determined by measuring the light absorbance at 260 nm (A_260_), and the ratio of A_260_/A_280_. reverse transcription was carried out in a 25 µL reaction volume with 2 µg of total RNA, according to the manufacturer’s protocol for Moloney-Murine Leukemia Virus (M-MLV) reverse transcriptase (Promega, Madison, WI, USA). Real-time PCR was performed using the ABI Prism 7900 Sequence Detection system (Applied Biosystems, Foster City, CA, USA). The cDNAs were amplified in triplicate with primers for human p21 and for GAPDH by using the Power-SYBR green mix with ROX (Applied Biosystems). Primer sequences were as follows: human p21 forward: 5’-TGAGCCGCGACTGTGATG-3’; p21 reverse: 5’-GTCTCGGTGACAAAGTCGAAGTT-3’; human GAPDH forward: 5’-TCGACAGTCAGCCGCATCT-3’; GAPDH reverse: 5’-CCGTTGACTCCGACCTTCA-3’. The PCRs were validated by the presence of a single peak in the melt curve analysis, and the amplification of a single specific product was further confirmed by electrophoresis on agarose gel. The human Oct-6 and catalase mRNA expression were analyzed by real-time PCR using specific TaqMan Gene Expression Assays conjugated with fluorochrome FAM, as follows: Oct-6 (Hs00538614_s1), catalase (Hs00156308_m1), and GAPDH (Hs03929097_g1) (Applied Biosystems). The relative expression of each gene versus the GAPDH was calculated according to the 2^−ΔΔCt^ method. The analysis was performed using the SDS version 2.4 software (Applied Biosystems).

### 4.9. Cell Cycle Analysis

After 48 h, the doxorubicin (0.5 µM)-treated H460 cells were washed in PBS with 0.1% sodium azide, and were fixed for 2 h at 4 °C in cold 70% ethanol. Thereafter, the cells were incubated for 30 min at room temperature with 50 µg/mL PI in PBS containing 0.5 mg/ml RNAse (Sigma-Aldrich, St. Louis, MO, USA), and were immediately analyzed using a FACSCanto (BD Bioscience, San Jose, CA, USA) and FlowJo (v7.2.5) Flow Cytometric Data Analysis Software (Tree Star, Inc., Ashland, OR, USA). 

### 4.10. Senescence-Associated β-Galactosidase (SA-β-Gal) Staining

The SA-β-Gal assay was performed to analyze the senescence of the H460 cells after 48 h treatment with 0.5 µM of Dox. The cells were left for 1 h with 100 nM bafilomycin A1 to induce lysosomal alkalinization, followed by 1 h of incubation with 33 µM of 5-dodecanoylaminofluorescein di-β-D-galactopyranoside (C_12_FDG; Invitrogen, Frederick, MD, USA). The percentage of SA-βGal positive cells was analyzed by using a FACSCanto (BD Bioscience, San Jose, CA, USA) and FlowJo (Version 7.2.5) Flow Cytometric Data Analysis Software (Tree Star, Inc., Ashland, OR, USA).

### 4.11. Immunofluorescence and Flow Cytometry

The expression of phosphorylated γH2AX in H460 cells treated with Dox for 48 h h was analyzed by immunofluorescence. For the intracellular staining, the cells were fixed with 1% formaldehyde, permeabilized with 70% ethanol, and then incubated with Fluorescein Isothiocyanate (FITC)-conjiugated-IgG1 or γH2AX. The fluorescence was analyzed using a FACSCanto (BD Bioscience, San Jose, CA, USA) and FlowJo (Version 7.2.5) Flow Cytometric Data Analysis Software (Tree Star, Inc., Ashland, OR, USA).

### 4.12. Statistics

The error bars represent the standard deviation (SD) or standard error of the mean (SEM). The data were evaluated by paired Student’s *t*-test.

## 5. Conclusions

Overall, our data show that the transcription factor Oct-6 is induced by genotoxic stress, and functions as a stress response effector in selected cancer cell lineages. In this context, the data shown in this manuscript further extend the knowledge of the complex range of actions mediated by DNA damage response, and underscore a possible role of Oct-6 in the protective response of cells to genotoxic oxidative stress. 

## Figures and Tables

**Figure 1 cancers-11-00810-f001:**
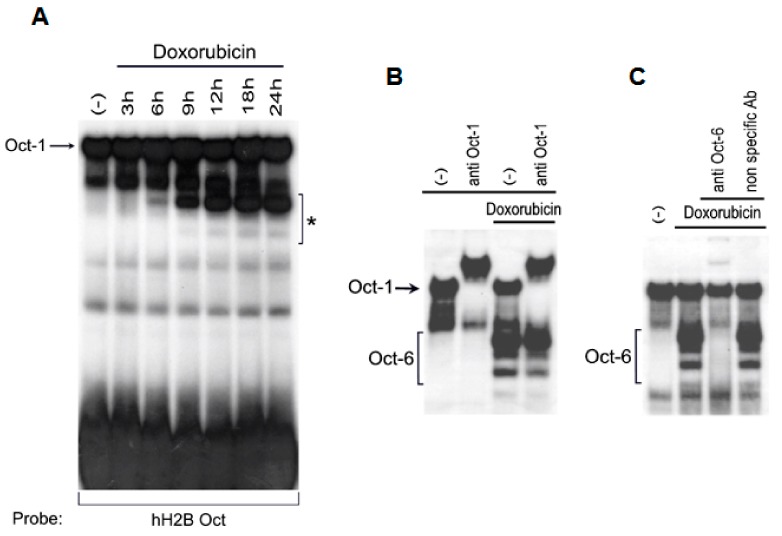
Doxorubicin (Dox) induces octamer DNA binding complexes in H460 cells. An electrophoretic mobility-shift assay (EMSA) was performed using the hH2B-octamer (Oct) sequence as a consensus octamer probe (**A**), in the presence of nuclear extracts (10 μg) from unstimulated (−) or Dox treated (0.5 µM) H460 cells for the indicated time. The asterisk indicates the position of the octamer binding complexes induced by Dox. (**B**,**C**) Supershift analysis of octamer binding complexes. Where indicated, anti-Oct-1 or anti-Oct-6 antibody or non-specific antibody was added to the reaction mixture. DNA binding complexes containing Oct-1 or Oct-6 are indicated in the figures. The EMSAs shown in the figure are representative of various independent experiments, all displaying similar results.

**Figure 2 cancers-11-00810-f002:**
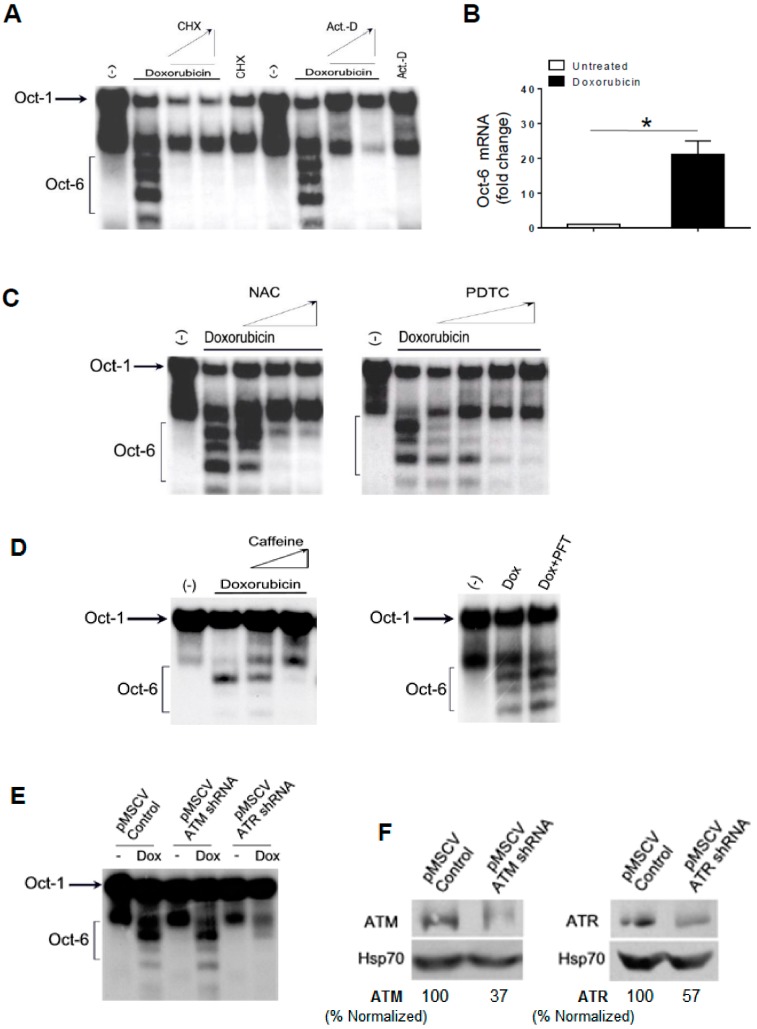
Doxorubicin induces Oct-6 in H460 cells: role of reactive oxygen species (ROS) and ataxia telangiectasia mutated- and Rad3-related (ATR). EMSA was performed, as described above, from H460 cells unstimulated (−) or treated with Dox for 24 h, in the absence or in the presence of different concentrations of 5 and 10 μg/mL of cycloheximide (CHX), or 1 and 5 μg/mL of actinomycin D (Act-D) (**A**), or N-acetylcysteine (NAC;10, 30, and 60 mM), or pyrrolidine dithiocarbamate (PDTC; 100, 200, 300, and 500 µM) (**C**), or caffeine (1 and 2.5 µM), or pifithrin-α (PFT; 30 μM) (**D**). The data are representative of one out of two independent experiments. (**B**) Real-time PCR analysis of the total mRNA obtained from H460 cells, untreated or treated with Dox (0.5 µM) for 24 h. The data, expressed as fold change units, were normalized with glyceraldehyde 3-phosphate dehydrogenase (GAPDH), and were referred to the untreated cells, considered as a calibrator, and represent the mean of three experiments (* *p* < 0.05). (**E**) EMSA was performed, as described above, in the presence of nuclear extracts (10 μg) from shRNA-ATM or ATR retrovirus infected H460 cells, unstimulated (−) or treated with Dox for 24 h. The data are representative of one out of two independent experiments. (**F**) Immunoblotting analysis for ataxia telangiectasia mutated (ATM) or ATR and Hsp70 of the total cellular proteins obtained from H460 cells infected with the empty control retrovirus (pMSCV control), or the retrovirus expressing ATM (pMSCV-ATM) or ATR (pMSCV-ATR) shRNA. The data are representative of one out of three independent experiments. The densitometric analysis of normalized ATM/Hsp70 and ATR/Hsp70 is shown (the whole blots are shown in the Appendix A).

**Figure 3 cancers-11-00810-f003:**
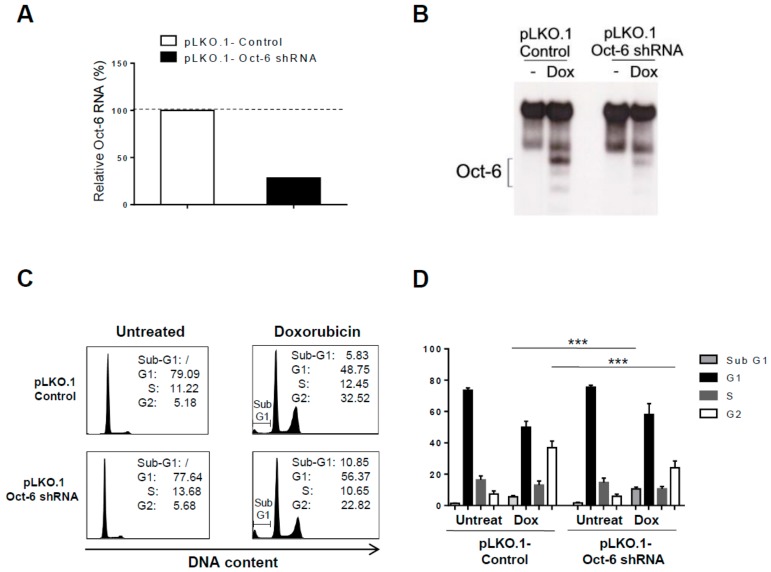
Knockdown of Oct-6 affects cellular response to genotoxic stress: effect on cell cycle arrest. (**A**) The total mRNA was obtained from untreated or 24 h Dox-treated H460 cells infected with lentivirus pLKO-shRNA-Oct-6 or non-target shRNA, and analyzed for Oct-6 mRNA expression by real-time PCR. The data, expressed as fold change units, were normalized with GAPDH, and were referred to the cells infected with non-target shRNA, considered as a calibrator, and represent the mean of three experiments (* *p* < 0.05; left panel). (**B**) EMSA was performed, as described above, in the presence of nuclear extracts obtained from pLKO-shRNA-Oct-6 or non-target shRNA infected H460 cells unstimulated (-), or treated with Dox for 24 h (right panel). (**C**) Control or Oct-6 silenced H460 cells, left untreated (-) or treated with Dox 0.5 μM for 48 h, were fixed and stained with propidium iodide (PI) to analyze the cell distribution among the different cell cycle phases. (**D**) Histograms of the different cell-cycle phases represent the mean of six independent experiments. Error bars indicate standard error of the mean (SEM; ** *p* < 0.01; paired *t*-test).

**Figure 4 cancers-11-00810-f004:**
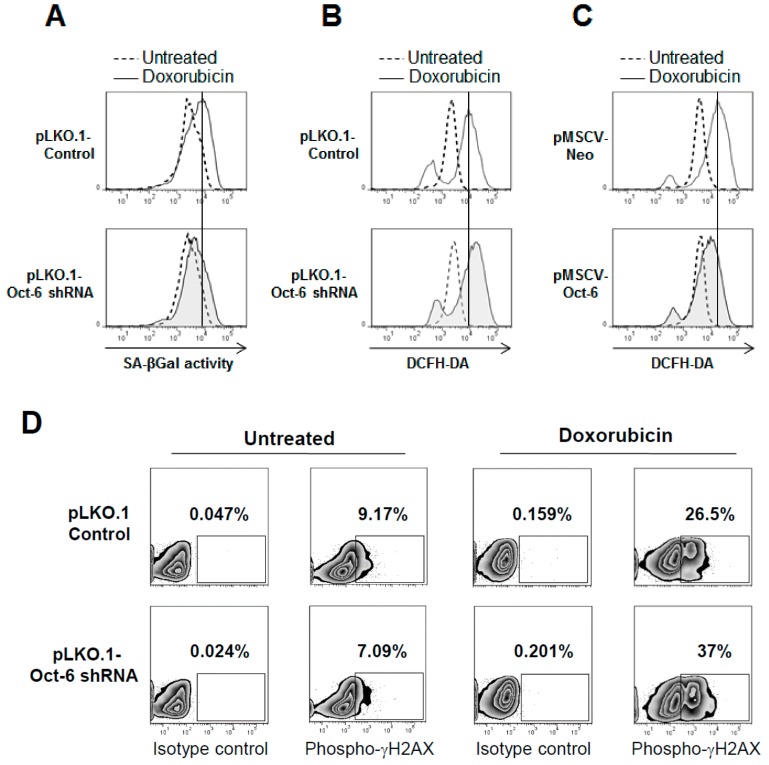
Oct-6 regulates drug-induced senescence, ROS generation, and DNA damage. (**A**) Control (black histogram) or Oct-6 silenced H460 cells (grey colored), left untreated (−) or treated with Dox 0.5 μM for 48 h, were left 1 h with 100 nM of bafilomycin A1 to induce lysosomal alkalization, followed by 1 h of incubation with C_12_FDG (33 mM), to analyze the percentage of senescence-associated *β*-galactosidase (SA-*β*Gal) positive cells by flow cytometry. (**B**) Intracellular ROS production was measured by staining pLKO-shRNA-Oct-6 (black histogram) or non-target shRNA infected H460 cells (grey colored) upon 48 h drug treatment with 10 μM DCFH-DA, as described in Materials and Methods, and analyzed by flow cytometry. (**C**) Intracellular ROS levels in H460 cells stably transduced with the retroviral expression vector pMSCV-Oct-6 or pMSCV-Neo unstimulated (grey colored), or treated with Dox (0.5 μM), (black histogram) for 48 h were evaluated by FACs analysis, as described above. The histograms shown in the figure are representative of various independent experiments, all displaying similar results. (**D**) The phosphorylation of γH2AX was evaluated by immunofluorescence and FACS analysis, by staining pLKO-shRNA-Oct-6 or non-target shRNA infected H460 cells with anti-γH2AX or anti-cIgG after 48 h exposure to Dox. All of the results shown are representative of one out of three independent experiments.

**Figure 5 cancers-11-00810-f005:**
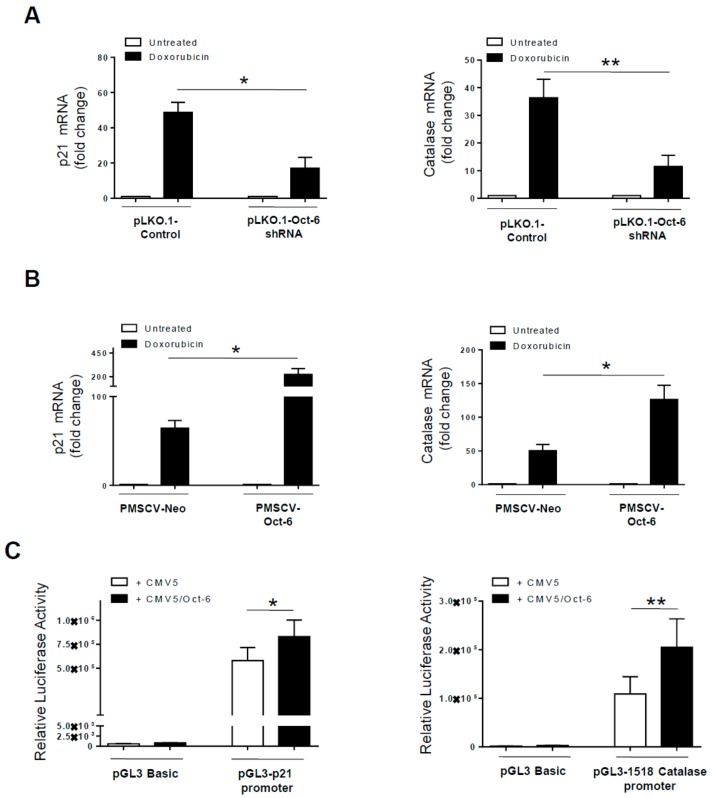
The p21 and catalase as Oct-6 target genes. The total mRNA was obtained from Oct-6 silenced or overexpressing H460 cells unstimulated (−) or treated with Dox for 24 h, and analyzed for p21 or catalase mRNA expression by real-time PCR (**A**,**B**). Data, expressed as fold change units, were normalized with GAPDH, and referred to the cells infected with non-target shRNA or pMSCV-Neo, considered as calibrators, and represent the mean of three experiments (* *p* < 0.05). (**C**) H460 cells were co-transfected with 5 μg of pGL3-Basic or p21 or catalase promoter Luc reporter vector, together with 2.5 μg of CMV5-Oct-6 construct (or CMV5 empty vector) and 1 μg of thymidine kinase (TK)-Renilla (or RSV-gal). Forty-eight hours after transfection, the cells were harvested, and the protein extracts were prepared for the luciferase assay. The results are expressed as relative luciferase activity normalized to protein concentration and to Renilla or *β*-galactosidase activity. Data represent the mean ± SEM from at least three experiments. (** *p* < 0.01; paired *t*-test).

**Figure 6 cancers-11-00810-f006:**
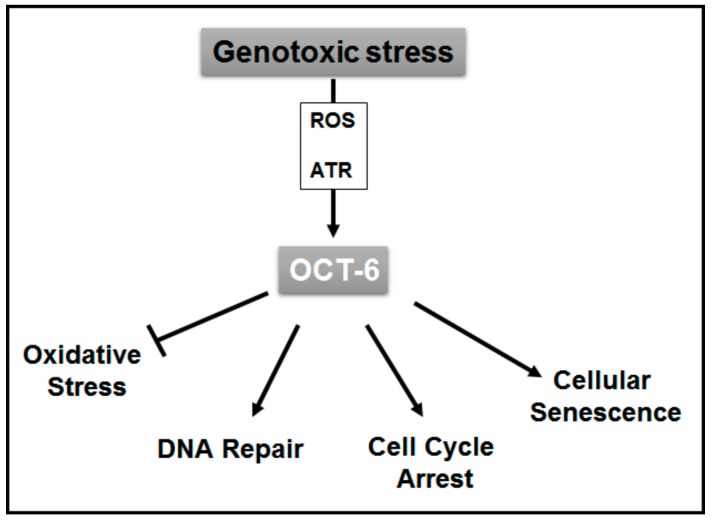
Proposed model for the role of Oct-6 in the regulation of cellular response to genotoxic stress. Genotoxic stress induces Oct-6 expression in cancer cells through ROS and ATR-dependent mechanisms. This transcription factor exerts a protective role against genotoxicity through the regulation of critical cellular processes triggered by DNA damage-mediated stress response. Oct-6 inhibits ROS production, decreases DNA damage, and promotes cell cycle arrest as well as cellular senescence.

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
