# Peer review of "The POU-Domain Transcription Factor Oct-6/POU3F1 as a Regulator of Cellular Response to Genotoxic Stress"

_cancers, 2019, doi:10.3390/cancers11060810_

Round 1

Reviewer 1 Report

COMMENTS TO THE AUTHOR(S)

Several lines of evidence in this study revealed that the transcription factor Oct-6 was involved in cellular response to genotoxic stress. The authors first found that expression of Oct-6 was induced by one of genotoxic drugs Dox in several cancer cell lines. Subsequently, they indicated that ROS production and also ATM/ATR activities were required for this upregulation. Loss of Oct-6 expression resulted in increased ROS and g-H2AX levels, and defect in cell cycle arrest at G2 phase after Dox treatment. These effects were seemingly due to downregulation of Catalase and p21. Finally, the authors confirmed that Oct-6 directly bound to the promoter region of p21. From these data, they concluded that Oct-6 functioned for the protective response of cells to genotoxic stress. I think this article is well-designed and should be published in cancers, however, I suggest minor revision before the editor(s) decide the final decision. My comments are as followed.

Lines 25-26: Why the authors focused on Oct-6 instead of other Oct transcription factors?

Line 36: the authors should insert “DNA damage response” (DDR).

Line 55: “γ"-radiation?  Doxorubicin (Dox) should be inserted.

Line 89: Did you check other Oct transcription factors? “electrophoretic mobility shift assay (EMSA)” should be inserted here.

Line 115: If Oct-6 binds to DNA with other proteins, we cannot conclude that fast migration in EMSA is due to truncation of Oct-6. Have you detected several bands of Oct-6 by the western blotting? And also I would like to known whether fast migrating complex is active or not.

Line 119 (Figure 2B): The intensity of the migrating band corresponding to Oct-1 complex should be reduced when Oct-6 was induced by Dox treatment, since DNA-binding of Oct-1 might be blocked by Oct-6.

Line 119 (Figure 2C): Where is the supershift band(s) of Oct-6 complex? Does the anti-Oct-6 antibody inhibit the DNA binding of Oct-6?

Line 150: Does the authors confirm whether Pifithrin-a inhibit p53 at this concentration?

Line 202: The readers might be confusing why the authors examine SA-b-galactosidase activity here. Please explain first.

Line 213 (Figure 3A-C): The authors should revise histograms. Lines of “untreated” are unclear.

Line 231: Did you examine expression of other genes known to be regulated by Oct transcription factors?

Line 250 (Figure 5): At the untreated condition, promoter activities in D were not coincided with mRNA levels in C. The authors should describe this difference.

Have you check the effect of Oct-1 as the control?

Line 315: I cannot agree with the description. If you want to claim that Oct-6 binds to Octamer consensus sequences of p21 promoter, you should do EMSA using fragments mutated in these sequences.

Author Response

Manuscript cancers-521252 by Fionda et al.

Dear Reviewer,

Thank you very much for the opportunity to resubmit a revised version of our manuscript “The POU-domain transcription factor Oct-6/POU3F1 as a regulator of cellular response to genotoxic stress” by Fionda et al., to be considered for publication in Cancers.

The comments and suggestions were highly valuable and have, to our opinion, clearly improved our manuscript.

We have addressed the comments as outlined in our point to point reply and have included data from additional experiments / analyses.

The manuscript has been amended accordingly and changes in the manuscript have been marked in yellow.

Please find attached the point-by-point responses letter.

Reviewer 2 Report

In this manuscript, “The POU-domain transcription factor Oct-6/POU3F1 as a regulator of cellular response to genotoxic stress” Fionda et al., demonstrated the Oct-6/POU3F1 is a novel modulator of genotoxic drugs to induce DNA damage-mediated stress response in cancer cells. These results also implied that Oct-6/POU3F1, previously recognized as a regulator of neural morphogenesis and epidermal differentiation, will be a target for the action of genotoxic drugs. Furthermore, the authors also provided detail mechanisms the induction of Oct-6 through ROS and ATR-dependent mechanisms, but in a p53 independent manner. These new findings will be interesting to the cancer research field and drive the new routes for the treatment of cancers. The manuscript was present clearly and easy to follow. Two comments are listed for reference:

1. To facilitate the reader to get a clear picture about the novel role for the Oct-6/POU3F1, a cartoon plot may be provided in the discussion section.

2. A major concern about this study is the all the results derived from cancer cell models, and major in the H460 NSCLC cells. I suggest that how to “translate” the results into the in vivo animal model should be discussed and this will make the manuscript more complete.

Author Response

Manuscript cancers-521252 by Fionda et al.

Dear Reviewer,

Thank you very much for the opportunity to resubmit a revised version of our manuscript “The POU-domain transcription factor Oct-6/POU3F1 as a regulator of cellular response to genotoxic stress” by Fionda et al., to be considered for publication in Cancers.

The comments and suggestions were highly valuable and have, to our opinion, clearly improved our manuscript.

We have addressed the comments as outlined in our point to point reply and have included data from additional experiments / analyses.

The manuscript has been amended accordingly and changes in the manuscript have been marked in yellow.

Please find attached the response letter.
